# Activity and Trafficking of Copper-Transporting ATPases in Tumor Development and Defense against Platinum-Based Drugs

**DOI:** 10.3390/cells8091080

**Published:** 2019-09-13

**Authors:** Raffaella Petruzzelli, Roman S. Polishchuk

**Affiliations:** Telethon Institute of Genetics and Medicine (TIGEM), Via Campi Flegrei 34, 80078 Pozzuoli (NA), Italy; r.petruzzelli@tigem.it

**Keywords:** ATP7A, ATP7B, cancer, membrane trafficking, copper homeostasis, cisplatin resistance, Golgi apparatus

## Abstract

Membrane trafficking pathways emanating from the Golgi regulate a wide range of cellular processes. One of these is the maintenance of copper (Cu) homeostasis operated by the Golgi-localized Cu-transporting ATPases ATP7A and ATP7B. At the Golgi, these proteins supply Cu to newly synthesized enzymes which use this metal as a cofactor to catalyze a number of vitally important biochemical reactions. However, in response to elevated Cu, the Golgi exports ATP7A/B to post-Golgi sites where they promote sequestration and efflux of excess Cu to limit its potential toxicity. Growing tumors actively consume Cu and employ ATP7A/B to regulate the availability of this metal for oncogenic enzymes such as LOX and LOX-like proteins, which confer higher invasiveness to malignant cells. Furthermore, ATP7A/B activity and trafficking allow tumor cells to detoxify platinum (Pt)-based drugs (like cisplatin), which are used for the chemotherapy of different solid tumors. Despite these noted activities of ATP7A/B that favor oncogenic processes, the mechanisms that regulate the expression and trafficking of Cu ATPases in malignant cells are far from being completely understood. This review summarizes current data on the role of ATP7A/B in the regulation of Cu and Pt metabolism in malignant cells and outlines questions and challenges that should be addressed to understand how ATP7A and ATP7B trafficking mechanisms might be targeted to counteract tumor development.

## 1. Introduction

The process of tumor development involves a remarkable rearrangement in the functions of virtually all cellular organelles. This includes the Golgi apparatus, which holds a strategic position at the crossroads of the main membrane trafficking pathways and harbors molecular platforms operating in intracellular signaling, cytoskeleton organization, and protein quality control mechanisms [1,2]. In malignant cells, the Golgi plays a significant role in the processing and focalized delivery of matrix digesting proteins that are required for invasion [3,4]. The Golgi also integrates membrane trafficking events with reorganization of cytoskeletal and cellular cues to modify the environment of the tumor cell and, hence, to promote its ability to conquer surrounding healthy tissue [2,5,6]. Finally, Golgi membranes host a variety of signaling molecules that control several checkpoints for mitotic progression, while in tumor cells the structural and molecular reorganization of the Golgi compartment frequently leads to a failure of these checkpoints and, as a result, to uncontrolled proliferation [7].

Over the past years, the wide spectrum of Golgi-associated oncogenic mechanisms expanded to include the regulation of copper (Cu) and platinum (Pt) metabolism in growing tumors. Cu operates as an essential cofactor for a variety of enzymes including those involved in cancer-driving processes [8]. Cu-dependent enzymes whose role in tumor growth and/or metastasis has been documented comprise mitogen-activated kinase 1 [9,10], superoxide dismutase 1 [11], cytochrome c oxidase [12], and the lysyl oxidase (LOX) group of proteins [13]. The discovery of elevated Cu levels in serum and tumors of cancer patients and their correlation with the gravity of the disease paved the way for the hypothesis that Cu supply to oncogenic enzymes represents a crucial process for tumor growth and metastasis [14,15,16]. This hypothesis gained further support from several reports showing the inhibitory impact of Cu chelators on tumor growth and malignancy in animal models as well as in patients [17,18,19]. By harboring the Cu-transporting ATPases ATP7A and ATP7B, the Golgi regulates the supply of Cu to several oncogenic metalloenzymes and the overall availability of Cu in tumor cells.

Additionally, the same Golgi-resident transporters ATP7A and ATP7B are also capable of pumping Pt across membranes rendering the tumor cells resistant to Pt-based cancer therapy [20,21,22]. The ability of cisplatin and similar Pt-based drugs to generate DNA adducts and, hence, to promote cell apoptosis has been used successfully to treat different cancers [23,24]. However, the efficacy of Pt-based therapy might be seriously compromised by the development of resistance mechanisms, which include ATP7A/B-mediated Pt excretion from tumor cells. Indeed, overexpression of ATP7A/B, their increased activity and accelerated trafficking towards Pt efflux/sequestration sites contribute to the tolerance of tumor cells to cisplatin and other Pt-based anticancer drugs [25,26,27]. Clinical studies documented that elevated expression of ATP7A and ATP7B correlates with poor outcome of cisplatin-based chemotherapy in patients with ovarian, endometrial, oral squamous cell, breast, gastric, esophageal, and lung carcinomas [28,29,30,31,32,33,34,35,36]. These findings prompted the idea of using ATP7A and B as predictive markers of Pt-based drug efficacy in solid tumors [25,26,27].

In this review, we discuss how the ability of the Golgi resident transporters ATP7A and ATP7B to handle Cu and Pt enables malignant cells to protect themselves from cisplatin-mediated death and to promote tumor growth and invasion.

## 2. Cellular Pathways Operating in Copper and Platinum Distribution

Cu represents an essential micronutrient that is required for the activity of enzymes driving fundamental cellular processes in living organisms, such as respiration, radical detoxification, iron uptake, neuronal development, and collagen remodeling [8,37]. The ability of Cu to shuttle between two oxidation states (Cu^1+^ and Cu^2+^) allows it to act as an enzymatic cofactor in several biochemical reactions [8,37]. Cu metabolism is finely tuned by several transporters and chaperone proteins to avoid toxic Cu accumulation in cells [8,37,38]. The import of Cu into cells is mediated mainly by the membrane protein Cu transporters 1 and 2 (CTR1, CTR2) [39,40,41]. After uptake, Cu chaperones distribute Cu to the various intracellular sites (Figure 1). CCS (Cu chaperone for superoxide dismutase) delivers Cu to the cytosolic enzyme superoxide dismutase 1 (SOD1), which detoxifies free superoxide radicals [42,43]. Another chaperone, COX17, supplies Cu to the cytochrome C oxidase complex in the mitochondria to support ATP synthesis [44,45]. Finally, ATOX1 ferries Cu to ATP7A and ATP7B in the trans-Golgi network (TGN) for the metalation of newly synthesized Cu-containing enzymes [46,47]. Despite an essential function in a number of vitally important processes, excess Cu poses a serious danger to the cell due to its ability to cause significant oxidative damage to different cellular components. Several cellular mechanisms are involved in the strict surveillance of Cu levels to avoid Cu toxicity [8,37,38]. Those include upregulation of metallothioneins and accelerated glutathione synthesis, which allow the cells to buffer increasing Cu levels [48,49]. In addition, an increase in Cu levels stimulates the trafficking of ATP7A/B from the Golgi to a number of post-Golgi sites, where ATP7A/B promote Cu sequestration and/or excretion [50,51].

A common feature of the proteins operating in Cu homeostasis is the presence of specific protein domains rich in cysteine, methionine or histidine, called metal-binding sites (MBSs). MBSs bind Cu^1+^ in a protective groove and convey the metal ion to the next acceptor protein through protein–protein interaction. This transfer mechanism allows Cu to be always bound to chaperones/transporters through labile electrostatic associations and avoids the presence of free highly reactive Cu in the cell [52,53]. Several lines of evidence suggest that Cu-binding proteins employ the same mechanisms to handle Pt. Key components of the Cu-transporting system such as CTR1, ATOX1, CSS, SOD1, COX17, and ATP7A/B have been shown to bind Pt [54,55,56,57,58,59]. It was also found that cells resistant to high levels of cisplatin also tolerate high Cu levels and vice versa [21]. Finally, changes in the level of expression of Cu transporters/chaperones influence cell resistance to the drug [25,26,27]. The correlation between Cu metabolism and cell resistance to Pt drugs led to the idea that modulating the activity of Cu transporters, including ATP7A/B, might be explored to counteract drug resistance and, therefore, to improve the efficacy of platinum-based cancer therapy.

## 3. The Physiological Role and Trafficking of the Copper-Transporting ATPases

Both ATP7A and ATP7B are able to transport Cu from the cytosol across membranes using ATP hydrolysis [60,61]. Mutations in genes encoding ATP7A and ATP7B cause two genetic disorders of Cu homeostasis, Menkes disease and Wilson disease, respectively [62,63,64,65]. The mammalian ATP7A and ATP7B proteins share high structural similarity (Figure 2). The N-terminal portion of both proteins contains six Cu-binding MBSs followed by eight transmembrane (TM) domains that form a channel to pump Cu from the cytosol. Nucleotide-binding (N) and phosphorylation (P) domains in the third cytosolic loop and an auto-phosphatase (A) domain in the second cytosolic loop regulate the ATPase activity of ATP7A/B [66,67,68].

The transfer of Cu by ATP7A/B across the membrane involves several steps. Acquisition of Cu by ATPases and the transfer of Cu to the transmembrane sites increase ATP7A/B affinity for ATP. The N-domain allows the selective binding of adenosine in ATP or ADP molecules [69]. After binding, ATP phosphorylates the aspartate residue in the highly conserved DKTG motif (Figure 2A) of the P-domain [70]. This “catalytic” phosphorylation of the aspartate causes conformational changes that allow Cu to be released from the transporter into the lumen of the secretory pathway. The crystal structures of prokaryotic orthologs of the ATP7A/B (bacterial or *Legionella pneumophilla* CopA) provided significant help for the modeling of these ATPases and understanding of their Cu-transporting activities [71,72]. A growing body of evidence has demonstrated that CPC stretch in the TM6 and the specific methionine in TM8 participate in Cu coordination within the membrane and, hence, are essential for Cu transport [71,72,73]. After Cu exits into the lumen of the secretory pathway, the ATP7A/B undergo further structural rearrangements, which allow hydrolysis of aspartate-phosphate bond by A-domain [74]. TGE motif in A-domain (Figure 2A) has been shown to play a key role in the auto-phosphatase activity of ATP7A/B [75,76]. The release of ADP and phosphate from their respective sites into the cytoplasm prepares ATP7A/B for a new cycle of ATP hydrolysis and Cu transport.

The structural identity, in turn, defines striking similarities in the function and trafficking of ATP7A and ATP7B. Both proteins normally reside in the TGN where they play a significant role in the biosynthesis of Cu-dependent enzymes. ATP7A/B transport Cu from the cytosol across the TGN membranes to the Golgi lumen where they load the metal onto newly synthesized Cu-dependent enzymes moving through the secretory pathway. The enzymes that receive Cu from ATP7A/B operate in connective tissue biogenesis (LOX) [77], neuropeptide processing (peptidylglycine alpha-amidating monooxygenase) [78], pigmentation (tyrosinase) [79], anti-oxidant defense (SOD3) [80], and in iron metabolism (hepcidin and ceruloplasmin) [81].

In response to increasing Cu, ATP7A/B traffic from the Golgi to post-Golgi compartments, where excess Cu might be transiently stored, and to the cell surface, where Cu excretion occurs [50,51,82,83,84]. Molecular analysis suggested that the trafficking of both proteins is likely to be mediated by a conformational change that promotes the phosphorylation of the aspartic acid in the motif DKTG of the P-domain with the release of ADP and the subsequent translocation of Cu through the channel formed by the eight transmembrane domains [75]. Indeed, mutations that affect aspartate phosphorylation block ATP7A/B in the Golgi [75].

Despite structural similarities, differential expression of ATP7A and ATP7B and targeting of these proteins to opposite domains in polarized cells define their main activities in maintaining Cu homeostasis at the level of the whole organism. This difference is underscored by the two genetic disorders, Menkes and Wilson diseases, which are caused by mutations in ATP7A and ATP7B, respectively, and lead to very different clinical manifestations [85,86]. ATP7A is expressed almost ubiquitously (with the exception of adult hepatocytes). However, ATP7A activity in the intestine constitutes the most critical step in Cu supply to the body. In enterocytes, incoming dietary Cu stimulates trafficking of ATP7A from the Golgi to the basolateral domain of the cells where ATP7A promotes Cu release into the portal circulation [87]. The DTAL motif at the end of the C-terminal tail has been shown to define basolateral targeting of ATP7A [88]. Mutations that affect expression of the ATP7A protein, its Cu-transporting activity or its ability to traffic from the Golgi do not allow Cu to be transported beyond the intestinal barrier. This results in severe Cu deficiency and, consequently, in dysfunction of Cu-dependent enzymes during Menkes disease pathogenesis, which manifests in extensive neurodegeneration, hypopigmentation and connective tissue abnormalities [86].

In contrast to ATP7A, ATP7B is mainly expressed in the liver, although lower ATP7B expression has been detected in other organs and tissues. In hepatocytes, ATP7B uses Cu coming from the portal circulation to metallate newly synthesized ceruloplasmin moving through the Golgi. After secretion into the blood, ceruloplasmin, which binds six Cu atoms, operates as a carrier that distributes Cu to peripheral organs and tissues [89]. A rise in Cu beyond safe intracellular levels promotes ATP7B export from the Golgi to endo-lysosomal compartments [84,90], where Cu might be stored transiently, and to the apical surface of hepatocytes where ATP7B promotes excretion of Cu to the biliary flow for further elimination of the excess metal from the body [82,83,84,91]. It has been shown that apical targeting of ATP7B in hepatocytes relies on the N-terminal signal ^37^FAFDNVGYE^45^ [83]. Disruption of ATP7B expression, activity and/or apical delivery causes Wilson disease with a strong buildup of Cu and subsequent hepatocyte damage, culminating in liver failure [66,85].

## 4. Copper-Binding Proteins and Multifactorial Mechanisms of Cisplatin Resistance

Although ATP7A and ATP7B are important effectors of Cu homeostasis in the body, overexpression of both proteins has been found to be associated with tumor resistance to cisplatin during chemotherapy [25,26,27]. Cisplatin (*cis*-diamminedichloroplatinum) is a front-line treatment against a variety of cancers such as testicular, ovarian, stomach, and bladder cancers [23,24]. Nowadays, next-generation cisplatin analogs, which include carboplatin and oxaliplatin, have been developed to increase the effectiveness of this anticancer drug against specific tumor cells while reducing overall cell toxicity [23,24]. The anticancer effects of cisplatin are exerted via intertwined signaling pathways [24,92]. Cisplatin itself is inert, but once inside the cell it undergoes hydrolysis producing a highly reactive platinum complex that is prone to interact with a wide number of intracellular substrates [24]. The complex binds DNA and triggers the formation of covalent Pt adducts that distort the helical structure of the DNA. Such DNA damage results in cell death via apoptotic or necrotic mechanisms [93].

Drug resistance is one of the major drawbacks in Pt-based cancer chemotherapy. Resistance exists in both acquired and intrinsic forms. The acquired form occurs when the drug is initially efficient but becomes ineffective over time, while the intrinsic form occurs when a drug is ineffective from the beginning of the treatment [24,94]. Of the two forms, the acquired one is the most problematic as it requires high drug doses to overcome the resistance and this often leads to an ineffective therapeutic outcome.

Different cisplatin resistance mechanisms have been discovered so far: inactivation of cisplatin by sulfur-containing molecules like glutathione and metallothionein, DNA repair, or reduced platinum accumulation by decreased drug uptake or increased drug efflux [24]. Cross-resistance of tumor cells to cisplatin and Cu indicates that the toxicity of both metals might be handled by the same molecular players [21]. Of these, glutathione and metallothionein play an important role in the development of resistance to chemotherapy. Glutathione forms conjugates with cisplatin through the activity of glutathione S-transferase that inactivates the drug by increasing its solubility and subsequent excretion by multidrug resistance-associated proteins (MRPs) [95,96]. Moreover, a recent study suggests that glutathione might be involved in the transfer of cisplatin to the ATOX1/ATP7B detoxification pathway [97]. Metallothioneins constitute a family of cysteine-rich low molecular weight proteins that bind to metals such as copper, zinc, cadmium, and mercury. These proteins act as detoxifiers of cisplatin in a variety of tumors, and overexpression of metallothioneins has been associated with a shorter survival rate after cisplatin therapy in esophageal and ovarian cancers [48].

Since the discovery of the connection between Cu homeostasis and the cellular response to cisplatin, experimental evidence suggests that the high-affinity Cu transporter CTR1 mediates the uptake of cisplatin into the cell [98,99]. Indeed, the expression of CTR1 was found to be reduced in some Pt-resistant cell lines, thereby contributing to the decrease in platinum accumulation [100]. Furthermore, knockdown of CTR1 has been shown to decrease cisplatin uptake by almost 80% in both yeast and mouse embryonic fibroblasts [98] thereby suggesting that reduced CTR1 expression is associated with tolerance to Pt-based chemotherapy. Correspondingly, low CTR1 expression was significantly associated with resistance to platinum-based chemotherapy and shorter survival of patients with ovarian cancer [100]. However, the precise mechanism for cisplatin uptake by CTR1 remains unknown. Since cisplatin has a larger size than Cu^1+^, it has been proposed that cisplatin may not be transported directly through the CTR1 channel. Rather, cisplatin may bind to extracellular motifs at the plasma membrane and trigger CTR1 internalization by endocytosis [54,101].

The Cu chaperone ATOX1 was also demonstrated to mediate cisplatin resistance using different mechanisms. Fibroblasts from Atox1-/- mice have been shown to decrease cisplatin uptake by preventing degradation of CTR1, which occurs in an Atox1-dependent manner and is stimulated by cisplatin [102]. Therefore, reduced ATOX1 expression levels in specific cancer cells might lead to cisplatin resistance. By contrast, elevated expression of ATOX1 has been associated with tolerance to cisplatin in several tumor cell lines [57]. Structural studies have demonstrated that cisplatin is able to bind to the CxxC Cu-binding motif in ATOX1 [56] thereby inducing unfolding of ATOX1 in vitro [56]. It is tempting to speculate that this cisplatin-mediated unfolding of ATOX1 might trap cisplatin in an ATOX1-bound state and prevent Pt from entering the nucleus to exert its toxic effects. Lastly, similarly to Cu, ATOX1 can bind and transfer cisplatin to the MBDs of ATP7B and this interaction represents an important pathway for cisplatin detoxification in the cell [59].

## 5. Trafficking and Activity of ATP7B and ATP7A in Cisplatin Resistance

Over the past two decades, numerous studies have indicated that overexpression of ATP7A/B in different tumor cell lines is associated with the development of resistance to both cisplatin and Cu with a reduced accumulation of the drug in transfected cells [21,103]. The discovery that cells selected for resistance to cisplatin are cross-resistant to Cu and vice versa suggests that cisplatin may enter the cell, be transferred within the cell, and be exported from the cell by transporters and chaperones that normally mediate Cu homeostasis. This hypothesis indicates that ATP7A/B might sequester the platinum drugs into subcellular compartments and/or promote their efflux from the cells, hence limiting Pt cytotoxicity [25,26,27].

Clinical observations revealed that patients with ATP7B-positive tumors had a significantly poorer response to chemotherapy as compared to patients with ATP7B-negative tumors [30]. Further studies showed that the levels of ATP7B were inversely correlated with cisplatin sensitivity in nine gynecological cancer cell lines (see example in Figure 3) and that ovarian cancer patients with ATP7B expression had a significantly poorer response to cisplatin-based chemotherapy than patients lacking detectable ATP7B expression [104]. The contribution of ATP7B to cisplatin resistance was also confirmed using a silencing approach. RNAi-mediated suppression of ATP7B enhanced the sensitivity of resistant ovarian cancer cells to cisplatin and promoted the ability of cisplatin to reduce the growth of tumor engrafts in nude mice [105]. In this context, ATP7B emerged as a predictive marker of chemoresistance for cisplatin-based regimens [100].

An implication of ATP7A in chemoresistance has also been investigated in human ovarian carcinoma cells where overexpression of this ATPase was able to confer resistance to Pt-based drugs [103]. Interestingly, this ATP7A-mediated resistance was found to be associated with an increase rather than a decrease of overall Pt levels in the cells, suggesting that overexpression of this Cu transporter could alter the intracellular distribution of active Pt molecules [103]. ATP7A overexpression also correlates with poor performance of cisplatin-based therapy in patients with ovarian and non-small cell lung cancers [28,106]. Recent in vivo studies further supported the role of ATP7A as a potential therapeutic target to prevent resistance to cisplatin. Indeed, deletion of ATP7A in tumorigenic fibroblasts was found to increase cisplatin sensitivity through an increased susceptibility to ROS and hypoxia, which substantially limit tumor growth in mice [107].

While clinical data clearly indicate a correlation between the levels of ATP7A/B and tolerance to Pt-based chemotherapy, the mechanisms by which ATP7A/B promote cell resistance remain to be fully understood. Although active transport of platinum molecules across the cell membrane by ATP7B has been suggested to be the primary mechanism of cisplatin resistance [108], the hypothesis that ATP7A/B might operate as specific transporters for cisplatin has been the subject of substantial criticism. This is because ATP7A/B are highly selective for Cu and the size and chemistry of Cu and cisplatin differ to a significant degree. Indeed, it has been proposed that cisplatin undergoes chemical transformation into much smaller Pt^2+^ prior to ATP7A/B-mediated translocation across the membrane [109]. Several lines of evidence suggest that Cu-ATPases operate as Pt transporters. In vitro studies demonstrated that accumulation of Pt in membrane vesicles requires ATP7B and its ATPase activity [108]. An ATP7B mutant variant with a disrupted transmembrane Cu-binding site (CPC to CPA mutation in the signature P1B-ATPase motif) was unable to catalyze Pt transport [108]. ATP7A has been also shown to support Pt transport across membranes in vitro [110]. Interaction of ATP7B with Pt requires CXXC motifs in the metal-binding domains, which are essential for the ability of ATP7B to control Pt transport, and thus to mediate resistance to Pt-based drugs [111]. Like Cu, cisplatin triggers the acyl phosphorylation of ATP7B and its relocation from the TGN to peripheral vesicles [111].

It turns out that resistance to cisplatin requires the ability of ATP7A/B to traffic from the Golgi in response to treatment with the drug. In the case of Cu, ATP7A/B trafficking from the Golgi allows the cells to sequester and excrete the excess metal, which is potentially toxic. This trafficking process is tightly regulated by conformational changes of ATP7A/B as well as by Cu-dependent interactions of these proteins with components of membrane trafficking machineries [68,112]. How ATP7A/B trafficking is coordinated with cisplatin detoxification in tumor cells remains poorly understood. Confocal imaging of a fluorescent cisplatin analog revealed its accumulation in peripheral ATP7B-positive vesicles [113], which are likely to be involved in Pt sequestration. Indeed, both ATP7A and ATP7B have been preferentially detected in vesicular structures of cisplatin-resistant cells, while sensitive cells exhibited both proteins, mainly in the Golgi [114]. However, the identity of these vesicular structures remains unclear. In hepatic cells, vesicular structures that receive ATP7B belong to the late endosomal (LE)/lysosomal compartment [84,90]. It is tempting to speculate that similar endo-lysosomal structures use ATP7B to sequester Pt in tumor cells. Indeed, the role of lysosomes in the sequestration and detoxification of Pt is well documented [115]. Considering that low pH stimulates the Pt-transporting activity of ATP7B [108], the acidic content of endo-lysosomal structures might generate a favorable environment for ATP7B-mediated import of Pt to their lumen. As a result, massive ATP7A/B-mediated sequestration of Pt in the lysosomes should reduce the probability of Pt-mediated DNA damage, thereby contributing to cisplatin resistance.

In hepatic cells, excess Cu might be transiently stored in ATP7B-positive LEs/lysosomes (to be released back in the case of Cu deficiency in the cytosol) or excreted into the bile through the exocytosis of these organelles [84,90]. Release of toxic Pt from the lysosomes to the cytosol would be dangerous for cellular homeostasis. Therefore, resistant cells should limit Pt storage in lysosomes by accelerating lysosomal exocytosis. A detailed analysis of the lysosomal system in cisplatin-tolerant ovarian cancer cells revealed a reduced number of lysosomes, which correlated with an elevated release of Pt-containing exosomes from these cells [116]. Taking into account that the release of exosomes occurs during lysosomal exocytosis [117], these findings suggest that activation of lysosomal exocytosis promotes elimination of Pt from the resistant cells.

Notably, exosomes from tumor cells have been shown to contain both ATP7A and ATP7B [116]. This finding might provide new insights into understanding how ATP7A/B handle cisplatin toxicity. Exosomes are derived from intraluminal vesicles (ILVs) residing in the interior of LE/lysosomal organelles [117]. Therefore, the presence of ATP7A and ATP7B in these structures suggests that both proteins traffic to LEs/lysosomes in Pt-resistant tumor cells. On the other hand, incorporation of ATP7A/B into exosomes suggests that ATP7A/B do not necessarily need to catalyze Pt transport across the membrane and might use different mechanisms to sequester Pt in the lumen of lysosomes. ILVs are precursors of exosomes and form via budding of LE/lysosome membranes towards the organelle lumen [117]. As a result, cytosolic portions of all transmembrane proteins, including ATP7A/B, end up inside the membrane-bound space of newly formed ILV structures. Therefore, binding of Pt to the cytosolic N-terminal MBSs of ATP7B would be sufficient for incorporation of the metal into ILVs (Figure 4), which would keep Pt in the lysosome until its exocytosis and concurrent exosome release. This scenario favors the idea that sequestration of cisplatin by MBSs of ATP7A/B, rather than active transport of Pt across the membrane, provides the key contribution to the resistance of tumor cells to cisplatin [92,109]. It is worth noting that hepatic cells keep ATP7B away from lysosomal ILVs/exosomes [84] and, thus, apparently do not use an exosome-mediated mechanism for sequestration and excretion of Cu. This indicates that Cu and cisplatin might have a different impact on lysosomal sorting of ATP7B. Moreover, tumor cells might sort ATP7B using molecular machineries/mechanisms that differ from those in hepatocytes and, hence, could be selectively targeted to suppress ATP7B-mediated resistance to cisplatin (see below).

## 6. Role of ATP7A in Activation of LOX-Dependent Oncogenic Mechanisms

LOX proteins have been identified recently as important effectors of tumorigenesis [13]. LOX is a secreted Cu-dependent amine oxidase belonging to the LOX family of five proteins (LOX, LOXL, LOXL2, LOXL3, and LOXL4) that share a conserved catalytic Cu-binding domain [118,119]. The LOX protein is synthesized as an N-glycosylated 48 kDa preproenzyme (preproLOX) which is then secreted out of cells as an inactive 50 kDa proenzyme (proLOX) that is subjected to proteolytic cleavage by the proteinase procollagen C to be converted into a mature form [118,119]. Activation of LOX also requires a supply of Cu, which is executed by ATP7A in the Golgi [77]. Once formed, the mature enzyme is able to induce collagen-elastin covalent cross-linking reactions within the ECM which results in an increase of ECM stabilization and tensile strength [118,119].

The ability of LOX to remodel the extracellular matrix promotes tumor cell migration and adhesion via the activation of focal adhesion kinase (FAK1) [120,121]. LOX and LOXL2 activities have been also shown to create a favorable environment for cell invasion and subsequent tumor growth at metastatic sites [120,122]. In addition to the well-known impact of LOX on the stability of the extracellular matrix, it has been shown that LOX is also involved in intracellular activities as the secreted protein is able to re-enter the cell and influence cell signaling [13,118,119]. Clinical data suggest that higher expression of LOX enzymes has been found to be associated with invasion, tumor growth, and metastasis in breast, colorectal, and ovarian cancers. Indeed, the expression of LOXL2 has been found to increase in invasive cancers and correlated with poor survival of patients [123]. Therefore, LOXL2 has been proposed as a novel marker for poor prognosis in distinct cancer types [124]. Elevated levels of LOXL4 have been found in lymph node metastases and in advanced tumor stages in head and neck squamous cell carcinomas [125].

Given the importance of LOX enzymes in cancer, there is an emerging effort to develop specific inhibitors of these enzymes for anticancer purposes. However, the development of LOX-specific inhibitors encountered several problems due to the lack of precise information regarding the 3D structure of these enzymes because of numerous difficulties in efficient LOX purification [126]. In the past, Cu chelators like D-penicillamine, tetrathiomolybdate, thiram and disulfiram have been used to block LOX activity in a non-specific manner, while b-aminoproprionitrile compounds were considered as irreversible LOX inhibitors [127]. The latter, however, were found to be toxic and therefore never used for therapeutic purposes [127]. The purification of the LOXL2 protein opened the possibility of developing more specific inhibitors that exhibited fairly promising efficacy in the treatment of fibrosis [128] and pancreatic cancer [129].

A recent study highlighted the possibility of using other approaches for the treatment of LOX-dependent malignancy. This strategy proposes to target ATP7A-mediated delivery of Cu to members of the LOX family within the Golgi compartments [130]. Indeed, silencing of ATP7A was able to inhibit LOX activity and, as a result, to suppress LOX-dependent mechanisms of tumor progression. ATP7A-deficient tumor cells exhibited a reduced ability to generate metastasis due to poor activation of FAK1 and low cell motility [130]. This suggests that ATP7A plays a key role in the activation of LOX proteins whose activities promote tumor progression and metastasis. In line with this notion, clinical data demonstrated that higher levels of ATP7A correlate with a reduced survival rate of patients with breast cancer [130]. Thus, targeting of ATP7A deserves serious consideration as a potential option for suppression of LOX-dependent oncogenic processes. It remains unclear whether elevated expression of ATP7B in certain tumors might favor LOX activities as well. Considering that ATP7A and ATP7B appear to be redundant in terms of activation of Cu-dependent enzymes, it would be important to test whether disruption of ATP7B function might also be used to inhibit LOX-mediated motility and invasiveness of malignant cells in ATP7B overexpressing tumors.

## 7. Prospective

Despite significant advances in investigating the role of ATP7A/B in cancer, understanding the mechanisms that drive their activity and trafficking in tumors lags behind our knowledge of the molecular networks regulating the function of these Cu transporters in Menkes and Wilson diseases. Considering that ATP7A/B function can facilitate tumor invasiveness and resistance to chemotherapy, many lessons from disease-causing ATP7A and ATP7B mutants could be learned to activate mechanisms that promote their loss of function in tumors. For example, a recent proteomics study revealed that the most frequent Wilson disease-causing mutant loses interactions with several trafficking proteins [131]. Suppression of these proteins in Pt-resistant tumor cells might affect ATP7B trafficking and hence tolerance to cisplatin chemotherapy. Importantly, the ATP7A interactome has been also characterized [132] and might be explored in a similar way to attenuate Pt-resistance and LOX activation in malignant cells. The use of ATP7A/B trafficking inhibitors in the tumors might be combined with Cu chelators. This strategy would both cut an ATP7A/B-mediated Cu supply to oncogenic Cu-binding proteins and impair ATP7A/B ability to confer cisplatin resistance through relocation towards Pt sequestration/efflux sites. Notably, a recent study demonstrates that Cu chelator tetrathiomolybdate inhibits platination of ATP7B via dimerization of MBSs [133], hence, rendering the protein less capable of driving Pt efflux from tumor cells.

Another important issue that has to be taken into account is that suppression of either ATP7A or ATP7B, which is desirable in tumors, might cause significant side effects in extra-tumor tissues. Considering that dysfunction of these transporters causes serious abnormalities of Cu homeostasis in Menkes and Wilson diseases, it would be extremely important to target ATP7A and ATP7B in a tumor-restricted manner. For this purpose, the mechanisms of ATP7A/B expression and trafficking in tumors have to be investigated to reveal differences from healthy tissues. These differences should be further explored for the development of therapeutic tools that would allow the activities of ATP7A/B to be disrupted specifically in tumors without affecting the function of the Cu-ATPases in other tissues. Our recent study suggests that this approach is feasible. By screening a library of FDA-approved drugs we found a few compounds that suppress ATP7B trafficking in Pt-resistant ovarian cancer cells and promote cisplatin toxicity. The same drug hits, however, did not have any significant impact on either ATP7B trafficking or cisplatin toxicity in hepatic cells, where ATP7B normally functions [134].

Finally, despite a significant contribution to tumor development and drug resistance, ATP7A and B do not work alone to trigger oncogenic processes. Indeed, cisplatin tolerance requires the orchestrated activities of at least two to three molecular pathways to promote the survival of tumor cells upon chemotherapy [24]. Therefore, when ATP7A/B activities in tumor cells cannot be targeted due to the risk of serious side effects, other mechanisms might be explored to circumvent ATP7A/B-mediated resistance to Pt-based drugs, especially considering that the list of such mechanisms continues to grow. For instance, autophagy has emerged recently as an important mechanism for Cu and Pt detoxification [135,136]. Loss of ATP7B and consequent Cu accumulation activate autophagy, which sequesters Cu-damaged components in ATP7B deficient hepatic cells [135], thus, preventing a propagation of Cu toxicity throughout different cellular compartments. Apparently, the similar autophagic defense mechanism might handle Pt-induced damage in tumor cells treated with cisplatin, hence, contributing to development of Pt-tolerance [136,137,138]. However, the potential of autophagy suppressors to promote Pt-toxicity in tumor cells with elevated ATP7A/B expression has yet to be tested. In this context, it is interesting to note that some drugs promoting Pt-cytotoxicity in such resistant cells downregulate expression of autophagy genes [134].

In conclusion, we believe that future studies of the molecular mechanisms regulating ATP7A/B activities, interactions, and trafficking in tumor cells represent a very challenging and, at the same time, very exciting task. The growing knowledge about the dysfunction of these proteins in Wilson and Menkes diseases should definitely help to determine how Cu and Pt metabolic pathways might be manipulated to suppress ATP7A/B-associated oncogenic processes. Thus, closing this gap in understanding ATP7A/B function in malignant cells will open new opportunities for the development and improvement of anti-cancer therapeutic strategies.

## Figures and Tables

**Figure 1 cells-08-01080-f001:**
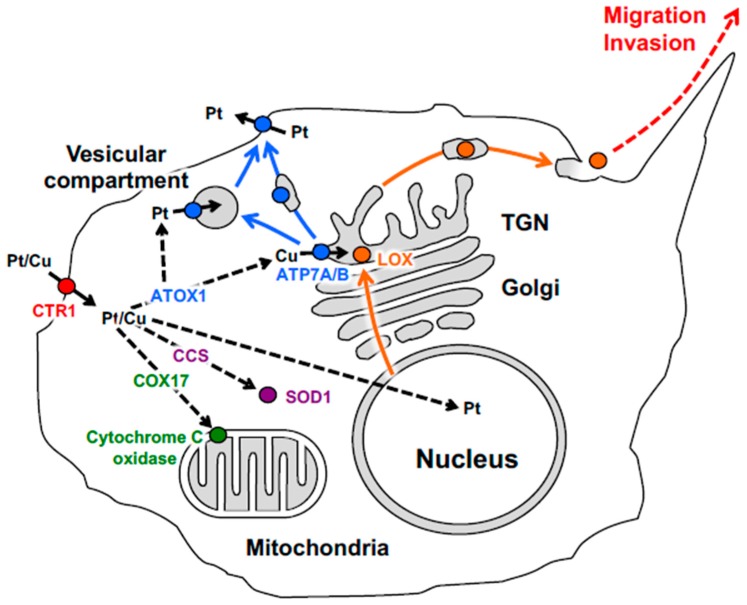
Schematic depiction of copper and platinum distribution in tumor cell. Copper (Cu) and cisplatin (Pt) taken up via CTR1 (red) are transferred to chaperones ATOX1, CCS and COX17, which ferry Cu or Pt derivates (black dash arrows) to ATP7A/B (blue) in the Golgi, to Cu–Zn superoxide dismutase (magenta) in the cytosol and to cytochrome c oxidase (green) in the mitochondria respectively. In the Golgi ATP7A/B load Cu on newly synthesized cuproenzyme lysyl oxidase (LOX, orange ball), which traffics along the biosynthetic pathway (orange arrows). Metallated LOX is secreted to the cell exterior (orange arrows), where it promotes migration and invasion (orange dash arrow) of the malignant cell. A significant increase in intracellular Pt induces export of ATP7A/B (blue arrow) towards vesicular compartment and cell membrane where ATP7A/B drives the sequestration and efflux of excessive Pt. Bold black arrows indicate Pt/Cu translocation across the membrane.

**Figure 2 cells-08-01080-f002:**
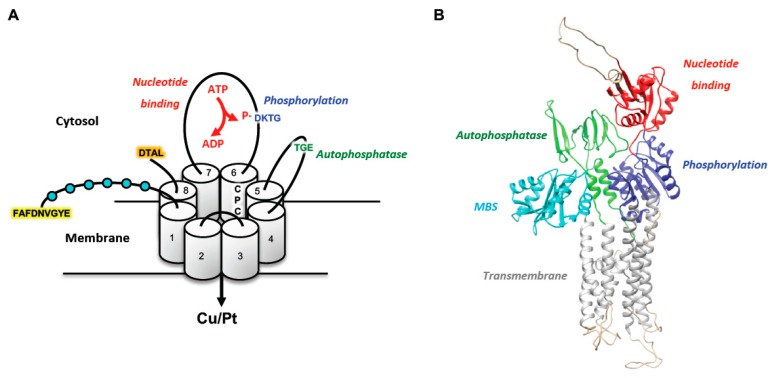
ATP7A/B structure. (**A**) Schematic structure of the ATP7A/B. The proteins have eight transmembrane (TM) domains, which form a pore for transport of Cu and Pt across the membrane. ATP7A/B contain ATP-binding (red), phosphatase (green), and phosphorylation (blue) domain, which regulate catalytic activity of the protein. N-terminal tail comprises six metal-binding sites (MBSs, depicted as cyan balls) that interact with Cu or Pt and regulate the protein conformation. In addition, the CPC motif in the sixth transmembrane domain plays a key role in the metal translocation along the channel. N-terminal apical sorting signal of ATP7B is highlighted in yellow, while basolateral targeting signal in C-tail of ATP7A is highlighted in orange. (**B**) Three-dimensional structures of ATP7B based on the existing structures of soluble domains and the previously determined structure of the homologous LpCopA from the bacterium *Legionella pneumophila*. Only MBSs 5 and 6 are shown in the model (cyan). Other domains of the protein are depicted with the same color as in panel A: ATP-binding (red), phosphatase (green), and phosphorylation (blue). The image was generated by UCSF Chimera software using the coordinate file deposited at the PUMPKIN website (http://mbg.au.dk/forskning/forskningscentre/pumpkin/atp7ab/).

**Figure 3 cells-08-01080-f003:**
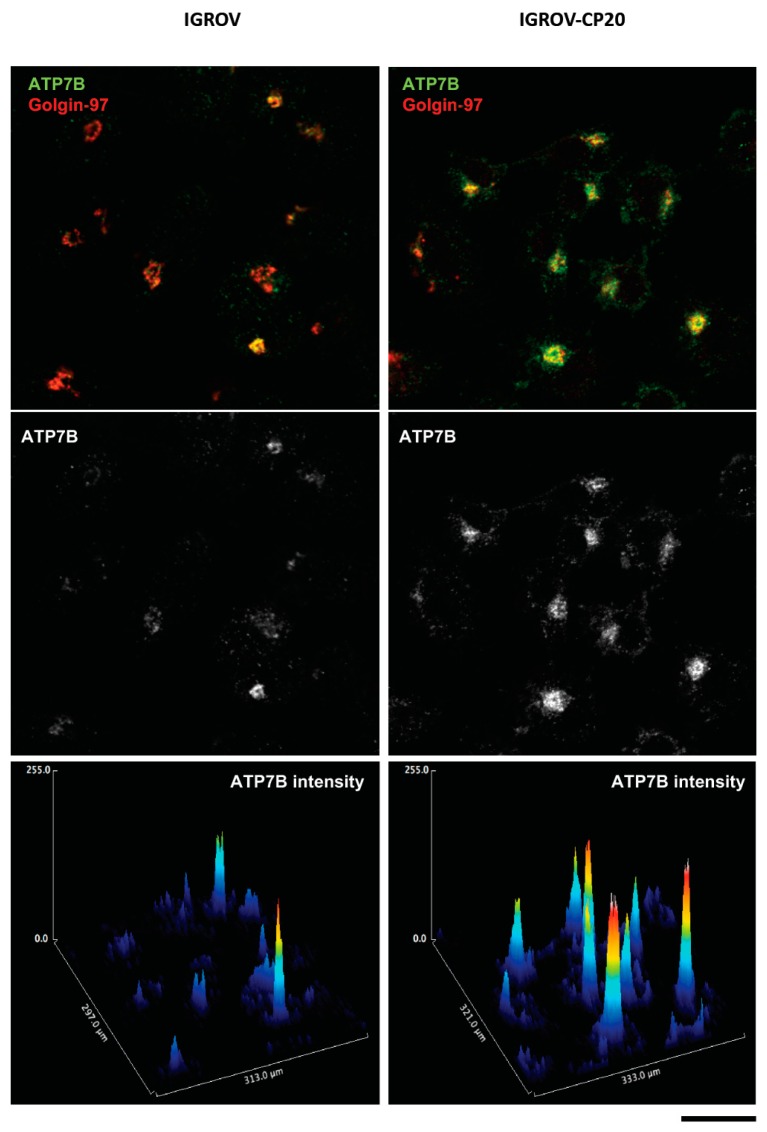
Expression of ATP7B in cisplatin-resistant and cisplatin-sensitive ovarian cancer cells. Pt-sensitive (IGROV) and Pt-resistant (IGROV-CP20) ovarian cancer cells were labeled for endogenous ATP7B and TGN resident protein Golgin-97 and then visualized using confocal microscopy. Graphs (bottom row) show intensities of ATP7B signal in corresponding images. Pt-resistant IGROV-CP20 cells exhibit higher levels of ATP7B compared to parental Pt-sensitive IGROV line. Scale bar: 25 µm.

**Figure 4 cells-08-01080-f004:**
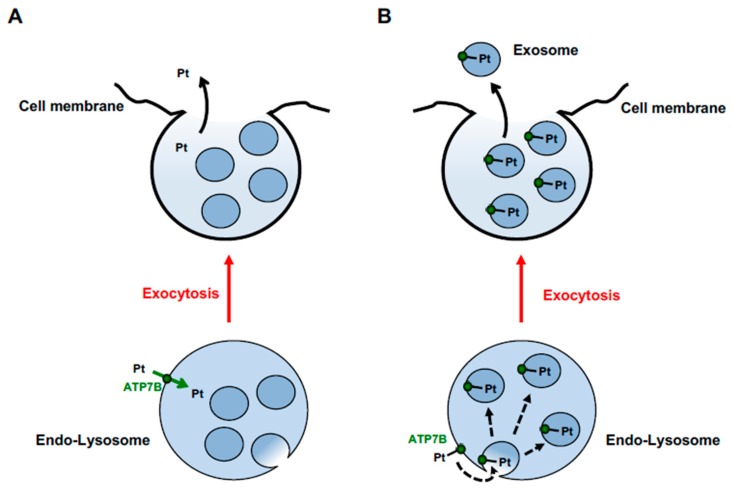
Hypothetical mechanisms of ATP7B-mediated Pt excretion through endo-lysosomal vesicular structures. (**A**) ATP7B transports Pt, or its derivates, from the cytosol to the lumen of vesicular endo-lysosomal structure across its membrane (green arrow). Pt is released from the lumen of vesicular structure (black arrow) during the process of its fusion with the plasma membrane (exocytosis, red arrow). (**B**) N-terminal MBSs of ATP7B bind Pt at the cytosolic surface of endo-lysosomal structures. ATP7B-bound Pt is sorted into intraluminal vesicles dash arrow budding towards the lumen of endo-lysosomal organelle. During exocytosis (red arrow) Pt-loaded intralumenal vesicles are released into the extracellular space as exosomes (black arrow).

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
