# Peer review of "Activity and Trafficking of Copper-Transporting ATPases in Tumor Development and Defense against Platinum-Based Drugs"

_cells, 2019, doi:10.3390/cells8091080_

Round 1

Reviewer 1 Report

In the review “Activity and trafficking of copper-transporting ATPases in tumor development and self-defense” Dr. Polishchuk and Dr. Petruzzelli provide a comprehensive overview of the recent findings and developments in the study of: 1) the physiological and pathological roles played by copper ATPases ATP7A and ATP7B in copper homeostasis 2) their roles in tumor development/ resistance mechanisms (with strong emphasis on Pt-based drugs).

Overall, the review is well written, easy to follow (considering the amount of details included in the discussion) and extensively referenced. The authors properly introduce the role of the Golgi trafficking pathways in relation to ATP7A/B physiology, discuss their involvement in cellular pathways responsible for Cu and Pt distribution and present an overview of the roles played by Cu-binding proteins in the mechanism of cisplatin resistance (with extended discussion on ATP7A/B). In addition, the authors also discuss in detail recent literature regarding the role of ATP7A in activation of LOX-dependent oncogenic mechanisms.

The only part that raise a concern is the very limited discussion about the molecular details and structural properties of ATP7A/B underlying its function as a copper transporter and putative Pt-binding/transporter protein.

A number of biochemical studies and molecular structures of a copper Cu+-transporting PIB-ATPase from Legionella pneumophila (LpCopA) in copper–free conformational states of its catalytic cycle have established the structural framework by which Cu-transporting P1B-type ATPases perform their function (Gourdon, Liu et al. Nature, 2011; Andersson, Mattle et al., Nat. Struct. Mol. Biol., 2014; Mattle et al. EMBO J, 2015). These and a number of other studies from the Lustsenko group have provide a large body of information concerning the mechanism responsible for substrate recognition and translocation by Cu+ pumps not discussed in the review. The manuscript would significantly benefit if these topics could be included. Also, the simplistic model presented in Figure 2 could be significantly improved, for example by presenting a more detailed 3D model for ATP7A/B which have been devised in more detail by Gourdon et al. (“Structural models of the human copper P-type ATPases ATP7A and ATP7B”, Biol. Chem., 2012).

After these changes will be introduced, I could recommend the manuscript for publication.

Author Response

Overall, the review is well written, easy to follow (considering the amount of details included in the discussion) and extensively referenced.

We would like to thank Reviewer for the positive remarks and helpful suggestions.

The only part that raise a concern is the very limited discussion about the molecular details and structural properties of ATP7A/B underlying its function as a copper transporter and putative Pt-binding/transporter protein.

A number of biochemical studies and molecular structures of a copper Cu+-transporting PIB-ATPase from Legionella pneumophila (LpCopA) in copper–free conformational states of its catalytic cycle have established the structural framework by which Cu-transporting P1B-type ATPases perform their function (Gourdon, Liu et al. Nature, 2011; Andersson, Mattle et al., Nat. Struct. Mol. Biol., 2014; Mattle et al. EMBO J, 2015). These and a number of other studies from the Lustsenko group have provide a large body of information concerning the mechanism responsible for substrate recognition and translocation by Cu+ pumps not discussed in the review. The manuscript would significantly benefit if these topics could be included. Also, the simplistic model presented in Figure 2 could be significantly improved, for example by presenting a more detailed 3D model for ATP7A/B which have been devised in more detail by Gourdon et al. (“Structural models of the human copper P-type ATPases ATP7A and ATP7B”, Biol. Chem., 2012).

 We expanded the text of the review according the suggestion of the Reviewer (pages 5-6) and included relevant literature.

Reviewer 2 Report

This a well balanced review on the role of ATP7A/B in cancer, with emphasis on the metabolism of Pt-based drugs. I have only minor comments.

-The authors should modify the title to highlight one of the major focus of this review, "Pt-based drugs".

-On Fig. 3. The magnification of the confocal image is too small. Nucleus is not labelled. The immunostaining for Golgin-97 is difficult to distinguish, specially on the bottom panel. No scale bar.

-On Fig. 4. ATP7B should be associated to exosomes.

-On the prospective. The authors should discuss combined therapies of copper-chelators and inhibitors of ATP7A/B trafficking to treat cancer.

-Expand the information that link autophagy, copper and cancer. The authors can add more specific information to this review. How do ATP7A/B regulate autophagy? How does copper regulate autophagy? How do Pt-based drugs interefere with autophagy? The authors can integrate this information on Fig. 4

Author Response

This a well balanced review on the role of ATP7A/B in cancer, with emphasis on the metabolism of Pt-based drugs. I have only minor comments.

We would like to thank Reviewer for the positive evaluation of our manuscript and helpful comments.

-The authors should modify the title to highlight one of the major focus of this review, "Pt-based drugs".

The title was modified

-On Fig. 3. The magnification of the confocal image is too small. Nucleus is not labelled. The immunostaining for Golgin-97 is difficult to distinguish, specially on the bottom panel. No scale bar.

We rearranged the figure to make the panels bigger in size. The main idea of the figure is to demonstrate a higher expression of ATP7B in Pt-resistant cells. For this purpose, a representative field with numerous cells is shown for each cell type. After making the images bigger, we hope the Reviewer could better appreciate the Golgin-97 labelling (red signal). We also agree that the higher ATP7B intensity in images from IGROV-CP20 cells might give an impression that Golgin-97 signal is weak or poorly defined. However, at the closer look it is quite clear that Golgin-97 staining substantially contributes to the yellow color in the Golgi area where overlap with ATP7B signal occurs. Finally, we also provided the scale bars as requested by the Reviewer.

-On Fig. 4. ATP7B should be associated to exosomes.

The figure was corrected according the Reviewer’s suggestion.

-On the prospective. The authors should discuss combined therapies of copper-chelators and inhibitors of ATP7A/B trafficking to treat cancer.

We added the discussion of a combinatory approach to prospective section (page 14).

-Expand the information that link autophagy, copper and cancer. The authors can add more specific information to this review. How do ATP7A/B regulate autophagy? How does copper regulate autophagy? How do Pt-based drugs interefere with autophagy? The authors can integrate this information on Fig. 4

 We expanded discussion on the link between autophagy, ATP7B and cisplatin resistance (page 15). We feel, though, that it would be difficult to integrate the information regarding autophagy on Fig. 4. Otherwise, it would be too busy.